# Abdominal Actinomycotic Abscess after Laparoscopic Sleeve Gastrectomy for Morbid Obesity: A Case Report

**DOI:** 10.3390/medicina59091516

**Published:** 2023-08-23

**Authors:** Ho-Goon Kim, Ho-Kyun Lee, Eunkyu Park

**Affiliations:** 1Department of General Surgery, Chonnam National University Medical School, 160, Baekseo-ro, Dong-gu, Gwangju 61469, Republic of Korea; dr4477@hanmail.net; 2Department of General Surgery, Chonnam National University Hospital, 42, Jebong-ro, Dong-gu, Gwangju 61469, Republic of Korea

**Keywords:** actinomycosis, laparoscopic sleeve gastrectomy, percutaneous drainage

## Abstract

Actinomycosis is a rare, chronic, suppurative, and granulomatous bacterial disease. The *Actinomyces* species exist as normal flora in the oropharynx, gastrointestinal tract, and the female genital tract. They are incapable of penetrating the normal mucous membranes and become pathogenic only when this barrier has been destroyed by trauma, surgery, immunosuppression, or after viscus perforation. We report the first case of an actinomycotic abscess after laparoscopic sleeve gastrectomy. A 29-year-old man underwent a laparoscopic sleeve gastrectomy with no intra-operative complications. On postoperative day 3, the patient had a fever with elevated inflammatory markers. Abdominal computerized tomography (CT) with oral water-soluble contrast media showed no extra-luminal leakage and no fluid collection adjacent to the resected stomach, other than the fluid collection in the right subhepatic space. Percutaneous drainage was attempted, but the procedure failed due to the patient’s thick abdominal wall. After two weeks of weight loss of about 12 kg, percutaneous drainage was successfully performed, and *A. odontolyticus* was identified through pus culture. After effective abscess drainage and high-dose antibiotics, the patient’s symptoms improved and the abscess pocket disappeared. We reported *Actinomyces* infection after gastric sleeve surgery. In the case of abscess formation after gastric sleeve surgery caused by actinomycete infection, antibiotic treatment and percutaneous drainage are effective together.

## 1. Introduction

Actinomycosis is a rare, chronic, suppurative, and granulomatous bacterial disease. In 1958, Batty isolated *Actinomyces odontolyticus* from patients with dental caries [1]. Since this report, 39 cases of infection caused by *A. odontolyticus* have been described. The *Actinomyces* species exist as normal flora of the oropharynx, gastrointestinal tract, and the female genital tract [2]. They are incapable of penetrating the normal mucous membranes and become pathogenic only when this barrier has been destroyed by trauma, surgery, immunosuppression, or after viscus perforation [3]. Actinomycosis is a disease caused by granulomatous inflammation defined by invasion, abscesses, and fistula development. In particular, only two cases of actinomycosis after bariatric surgery have been reported after bypass [4,5]. Herein, we report the first case of an actinomycotic abscess after laparoscopic sleeve gastrectomy.

## 2. Case Presentation

A 29-year-old man (136.5 kg, 186.0 cm, body mass index (BMI) 39.3 kg/m^2^) was admitted for bariatric surgery. The patient had a history of treatment for paranoid schizophrenia 10 years ago and severe sleep apnea, as well as a surgical history of septoplasty for sleep apnea. Social and familial histories were non-specific. Furthermore, the patient had no history of diseases or medication that could potentially cause immunosuppression. Preoperative chest posteroanterior (PA), gastroduodenoscopy, abdominal computed tomography (CT), and laboratory testing were carried out 2 weeks before surgery. The patient had no history of gallbladder stones, but due to limitations in physical examination and abdominal ultrasonography caused by obesity, an abdominal computed tomography (CT) scan was performed to differentiate gallbladder stones. No remarkable abnormal findings were observed on endoscopy and chest PA. Additionally, no evidence of hepatic mass and perihepatic abscess or inflammation was observed on the CT, apart from severe fatty liver findings (Figure 1a,b). In March 2020, a laparoscopic sleeve gastrectomy was performed with no intra-operative complications, and the surgical site was not irrigated. Routine prophylactic administration of 1st-generation cephalosporin was given before surgery and within 24 h after surgery.

On postoperative day 3, the patient had a fever of 38 °C, WBC count increased to 14.3 × 10^3^/μL, and the C-reactive protein level increased to 23.7 mg/dL. The following day, abdominal CT with oral water-soluble contrast media was performed, which showed a 50 × 40 mm sized abnormal fluid collection in the right subhepatic space. Extra-luminal leakage was not observed (Figure 2b) and apart from the fluid collection in the right subhepatic space (Figure 2a), there was no fluid collection adjacent to the resected stomach. Percutaneous drainage was attempted, but the procedure failed as the needle did not reach the fluid due to the patient’s thick abdominal wall. We decided to observe the patient without the use of antibiotics.

Subsequently, although the patient did not develop a fever, continued right upper abdominal mild tenderness was reported. On postoperative day 17, a follow-up abdominal CT scan was performed. which revealed a marked increase in multiple fluid collections in the right subhepatic space, measuring 135 × 105 mm, with the formation of an abscess wall, abut to the transverse colon. In addition, there was secondary inflammatory change in the colon (Figure 3a,b). At this time, the patient’s weight had reduced to 124 kg (%TWL: 9.2%, %EWL; 25.0%) and percutaneous drainage could be performed. In the pus culture, *A. odontolyticus* was identified. With a diagnosis of *A. odontolyticus*, and in consultation with the division of infectious disease, intravenous administration of 6.75 g of tazoperan per 8 h was initiated. A high dose of antibiotics was administered due to the patient’s obesity.

The abscess was effectively drained, and the right upper abdominal tenderness improved after 3 days (Figure 4a,b). However, a severe cough developed, and percutaneous drainage was performed for right pleural effusion. Chest CT was performed the following day as the cough did not sufficiently improve. Chest CT showed that the subhepatic abscess had markedly reduced, but the pleural effusion was multiloculated and had not been sufficiently drained. We drained the remaining pleural effusion through chest tubing.

On postoperative day 40 (24 days after percutaneous drainage), all symptoms had improved and the patient was discharged. The patient was 114.0 kg with a BMI of 33.0 kg/m^2^ (%TWL: 16.5%, %EWL; 45.0%). Six months after surgery, the patient’s weight had decreased to 103 kg and the BMI was 30 kg/m^2^ (%TWL: 24.5%, %EWL; 67.0%). No specific symptoms were observed in the patient, and no abscesses or other abnormalities were detected on follow up CT.

## 3. Discussion

Abdominal actinomycosis is a rare disease. Abdominal infections caused by *A. odontolyticus*, which was first isolated from the site of dental caries by Batty in 1958 [1], are extremely rare. We found only four cases of abdominal infection caused by *A. odontolyticus* in a Medline literature search [6,7,8,9]. Infections caused by *A. odontolyticus* tend to exhibit pulmonary effects [10,11,12]. This report is the first case of an abdominal infection caused by *A. odontolyticus*, which developed after bariatric surgery. Two case reports of *Actinomyces* infections after bypass surgery in patients with morbid obesity were reportedly caused by *A. Israelii* [4,5]. Moreover, the present report is the first case, to our knowledge, in which an abdominal actinomycotic abscess developed after gastric sleeve surgery.

There are several methods to confirm the microbiological diagnosis of actinomycosis. The first method is diagnosis by identifying the actinomycosis bacteria at the sterile site. However, the failure rate of bacterial cultures is high due to the previous use of antibiotics, inhibition of *Actinomyces* growth by concomitant and/or contaminant microorganisms, inadequate culture conditions, or short-term incubation [13]. The second method includes molecular techniques for identification via 16S ribosomal RNA (rRNA) sequencing [14]. Additionally, matrix-assisted laser desorption ionization time-of-flight mass spectrometry (MALDI-TOF MS) is a precise and efficient method for identifying *Actinomyces* [15]. The MALDI-TOF technique uses a matrix that absorbs laser energy to generate ions from large molecules, while minimizing fragmentation. [16]. In this case, the identification of *A. odontolyticus* was performed using MALDI-TOF MS.

Outside of the intestine, *Actinomyces* usually grows through local spread. In rare cases, it is spread through blood circulation, but not by lymphatic dissemination [17]. Abdominal actinomycosis usually develops after trauma, perforation of the intestine, endoscopic manipulation, or surgery of the gastrointestinal tract due to the spread of naturally occurring *Actinomyces* species into the peritoneal cavity. Acute perforated appendicitis and diverticulitis are common predisposing events [18,19,20].

In the case of the reported *Actinomyces* infection after laparoscopic bypass surgery, it is believed that the *Actinomyces* had grown on the mesh employed to avoid gastric dilatation [4]. In the present case, we observed an *Actinomyces* infection in a case of laparoscopic sleeve gastrectomy, where the remnant gastric mucosal barrier had not been opened during the procedure. *Actinomyces odontolyticus* was initially isolated from the deep carious dentine by Batty [1]. These organisms normally exist as part of the normal flora in the oropharynx, gastrointestinal tract, and the female genital tract. However, infections usually develop after instances of trauma, intestinal perforation, endoscopic manipulation, or gastrointestinal tract surgery. In the present case, in the process of removing the resected stomach through the 12 mm umbilical port site, resected stomach was forcibly extracted through a deep and narrow hole without the use of a laparoscopic bag. We predict that during this process, the pressure in the resected gastric lumen increased, causing micro-perforations in the stomach wall. Therefore, we believe that this abdominal infection originated not directly from the bacteria present in the teeth, but rather from *Actinomyces* that are part of the normal flora in the gastrointestinal tract. These *Actinomyces* likely spread into the peritoneal cavity through the area where mucosal barrier was broken. The selection of the stapler based on the thickness of the stomach wall will indeed have a significant impact on the leakage from the stapling line. In this case, for the initial stapling during sleeve gastrectomy, the Signia™ Stapling System with Endo GIA™ Black Reload was used, and for the remaining stapling, Endo GIA™ Purple Reloads were used according to the general laparoscopic sleeve gastrectomy method. As can be seen, CT performed on the 4th postoperative day showed no leakage of contrast media (Figure 2b). Additionally, CT performed on the 18th postoperative day showed the site where the abscess occurred, which is also the space around the liver that is at a distance from the stapling line and where the stomach was inadvertently left, supporting this possibility (Figure 3a,b). Furthermore, the lack of preoperative oral disinfection suggests that *Actinomyces*, which is part of the normal flora in the oral cavity and gastrointestinal tract, might have been given an opportunity to cause the infection, further supporting this possibility.

According to Massimo et al. [21], the use of prophylactic antibiotics in relation to obesity varies depending on whether they are hydrophilic or lipophilic. For cephazolin, high doses are unnecessary, and given as a single 2 g IV bolus 3–5 min before skin incision maintains protective cefazolin concentration for 4.8 h. In this patient’s case, 2 g of cephazolin was administered just before the skin incision, and the surgery lasted for 2.5 h, making the prophylactic antibiotic use appropriate.

In this case, the patient complained of mild heartburn and was prescribed a proton pump inhibitor (PPI) after the surgery. Therefore, it is important to consider the possibility of changes in the environment caused by the decrease in acid secretion due to PPI usage and its potential relation to actinomycosis infection. However, in this case, the abscess occurred in the subhepatic area, which is separate from the remnant stomach. As a result, it is believed that there is no direct correlation between the decrease in acid secretion due to PPI usage and the development of actinomycosis.

*Actinomyces* causes a granulomatous inflammatory reaction with abscess formation, followed by necrosis and extensive reactive fibrosis [22]. *Actinomyces* infections are susceptible to antibiotics [23] with penicillin being the treatment of choice for actinomycosis. Furthermore, aggressive surgical procedures are generally not recommended in these cases. Drainage procedures are usually performed for culture-assisted diagnosis [24]; however, in our experience, percutaneous drainage led to rapid clinical improvement, as well as a diagnosis. In addition, even large pyogenic liver abscesses can be successfully treated with percutaneous drainage [25]. Therefore, even if diagnosed as an *Actinomyces* infection, active percutaneous drainage is effective if abscess formation occurs. 

“Sourced control”, which refers to a set of measures aimed at damaging the host antimicrobial defense and allowing an organism to restore homeostasis, involves controlling the focus of infection and modifying factors in the infectious environment that promote microbial growth. As in this case, in healthy patients with no or well-controlled comorbidities and no immunocompromise, where the infection is the main problem, the goal should be to go beyond simply removing heavily contaminated material and focus on restoring optimal physiological function [26]. This demonstrates that in this case, the simultaneous use of antibiotics and percutaneous drainage is appropriate and effective.

## 4. Conclusions

Here, we reported an extremely rare Actinomyces infection after gastric sleeve surgery. The differential diagnosis of intraabdominal abscess after gastric sleeve surgery should include actinomycosis. In addition, in the case of abscess formation after gastric sleeve surgery caused by Actinomyces infection, antibiotic therapy and percutaneous drainage are effective together.

## Figures and Tables

**Figure 1 medicina-59-01516-f001:**
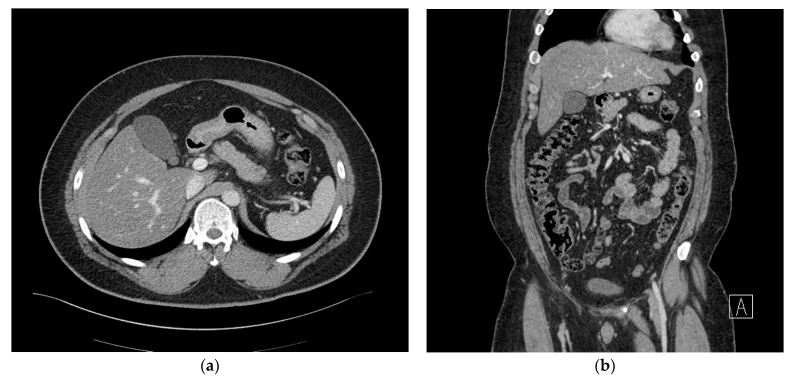
(**a**) Axial view of the abdominal computed tomography (CT) scan and (**b**) coronal view of the abdominal computed tomography (CT) scan showing no evidence of hepatic mass or perihepatic inflammation.

**Figure 2 medicina-59-01516-f002:**
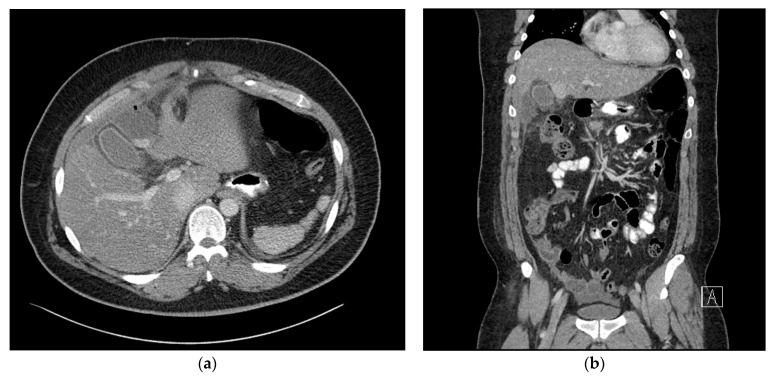
(**a**) Axial view of the abdominal computed tomography (CT) scan at the subhepatic level, showing a 50 × 40 mm sized abnormal fluid collection in the right subhepatic space. (**b**) Coronal view of the abdominal computed tomography (CT) scan showing no extraluminal leakage.

**Figure 3 medicina-59-01516-f003:**
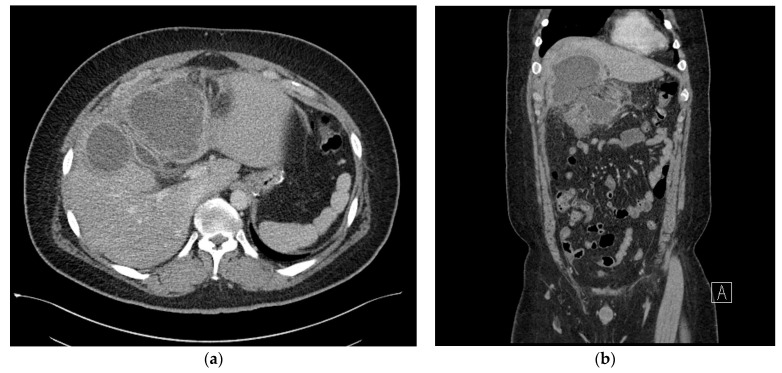
(**a**) Axial view of the abdominal computed tomography (CT) scan showing a marked increase in multiloculated fluid collections in the right subhepatic space. (**b**) Coronal view of the abdominal computed tomography (CT) scan showing multiloculated fluid collections in the right subhepatic space, abutting the transverse colon.

**Figure 4 medicina-59-01516-f004:**
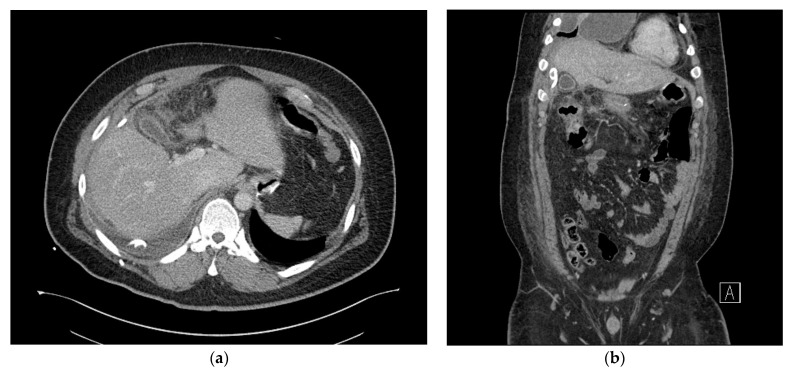
(**a**) Axial view of the abdominal computed tomography (CT) scan showing the multiloculated fluid collection in the subhepatic space showing significant improvement. (**b**) Coronal view of abdominal computed tomography (CT) scan demonstrating improvement in the adjacent transverse colon wall thickening, which was associated with inflammatory changes.

## Data Availability

Data are available from the corresponding author on reasonable request.

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
