# Peer review of "Abdominal Actinomycotic Abscess after Laparoscopic Sleeve Gastrectomy for Morbid Obesity: A Case Report"

_medicina, 2023, doi:10.3390/medicina59091516_

Round 1

Reviewer 1 Report

agree that this care represents a rare cause of hepatic abscess formation

given the size of the abscess on imaging, i find it hard to believe this formed in just two days after a sleeve, specially when there is no path of communication between these organs, abscesses usally take 5-7 days for form a wall. There is no mention of in immunocompromised status on the patient, and in an otherwise healthy young male i cant establish the causality between the gastrectomy and this rare abscess. 

why did this patient undergo preoperative CTAP - this is not part of any routine workup for bariatric surgery, specially if undergoing a sleeve gastrectomy? the gallbladder on the CTAP looks very abnormal, thickened wall, pericholecystic fluid, wonder if he had cholecystitis instead. how much time was in between the CT and the surgery? are we sure this abscess was not present preoperatively? images for this should be included as well 

also, authors should better profread the manuscript prior to submission... Line 99-114 show a "copy paste" section on the materials and methods guidelines from the journal.... 

Authors fail to establish or discuss a plausible connection or causality between gastric surgery and this abscess. although the stomach is entered and by definition this represents a clean contaminated case, there is usually no spillage of gastrointestinal contents following the standard stapling of the stomach... not sure "microperforations" when extracting the stomach are an issue / cause for this abscess. 

Author Response

Response to Reviewer 1 Comments

I am grateful for the comments and suggestions from you, and have responded to them and revised the paper according to your critiques.

Point 1: Given the size of the abscess on imaging, I find it hard to believe this formed in just two days after a sleeve, specially when there is no path of communication between these organs, abscesses usually take 5-7 days for form a wall. There is no mention of in immunocompromised status on the patient, and in an otherwise healthy young male I can’t establish the causality between the gastrectomy and this rare abscess. 

Response 1: I agree with your observation that it is unlikely to form a large abscess with an abscess wall in a short period of time after the surgery. In this particular case, the subhepatic fluid collection seen in Figure 1 (Figure 2 after modification) initially presented as an approximately 50 x 40 mm abnormal fluid collection without a discernible wall. Over the course of an additional two weeks, as depicted in Figure 2 (Figure 3 after modification), it progressed into an abscess measuring approximately 135 x 105 mm, accompanied by the formation of an abscess wall.

In response to your feedback, we have additionally described that the patient does not have an immunocompromised status (line 44–45), and have specified the size of the abscesses (line 63-66, line 78–81).

Point 2: why did this patient undergo preoperative CTAP - this is not part of any routine workup for bariatric surgery, specially if undergoing a sleeve gastrectomy? the gallbladder on the CTAP looks very abnormal, thickened wall, pericholecystic fluid, wonder if he had cholecystitis instead. how much time was in between the CT and the surgery? are we sure this abscess was not present preoperatively? images for this should be included as well

Response 2: Due to the limitations in the physical examination methods and abdominal ultrasound in detecting gallbladder stones in this patient with obesity, an abdominopelvic CT scan was performed to differentiate gallbladder stones. In this case, an abdominopelvic CT was conducted 14 days prior to surgery, which confirmed the absence of cholecystitis or surrounding inflammation.

In accordance with your feedback, we have included this image in the figure and described the absence of inflammation and abscesses around the gallbladder preoperatively. (line 50–53)

Point 3: also, authors should better proofread the manuscript prior to submission... Line 99-114 show a "copy paste" section on the materials and methods guidelines from the journal....

Response 3: I appreciate your insightful feedback regarding this error. I have taken the necessary steps to rectify the error by removing the contents of the template file, which inadvertently got included during the process of aligning with the format of Medicina.

Point 4: Authors fail to establish or discuss a plausible connection or causality between gastric surgery and this abscess. although the stomach is entered and by definition this represents a clean contaminated case, there is usually no spillage of gastrointestinal contents following the standard stapling of the stomach... not sure "microperforations" when extracting the stomach are an issue / cause for this abscess.

Response 4: In this case, the sleeve gastrectomy was performed by a surgeon with relatively limited experience. The surgeon attempted to remove the resected stomach without a plastic bag through the umbilical port site, where a 12 mm port had been inserted. However, during this process, it was observed intraoperatively that the serosa was torn and there were some perforations due to the pressure applied to the resected stomach. We believe that the subsequent formation of an abscess in the subhepatic space, where the stomach was inadvertently left, serves as indirect evidence of abscess formation resulting from this complication. (line 151–170)

Reviewer 2 Report

Authors described a rare post operative case of sleeve gastrectomy which developed intraperitoneal actinomycotic abscess post operatively. The causative bacteria A.odontolyticus is oral or gastrointestinal indigenous one, therefore the infection opportunity to peritoneal cavity seemed to be the gastric incision at the  sleeve gastrectomy.

The case was rare and worthy of report, and authors were recommended to describe more in detail as below.

1.  What devise did authors use at sleeve gastrectomy? and which size of staple for the resection?

2.  Which method of the oral care did authors do preoperatively? Did authors use oral disinfectant?

3. Authors were recommended the infectious route of A.odontolyticus of the case considering above points. 

Author Response

Response to Reviewer 2 Comments

I appreciate the valuable feedback from you, and I have made the necessary revisions to the paper in accordance with your suggestions.

Authors described a rare post operative case of sleeve gastrectomy which developed intraperitoneal actinomycotic abscess post operatively. The causative bacteria A.odontolyticus is oral or gastrointestinal indigenous one, therefore the infection opportunity to peritoneal cavity seemed to be the gastric incision at the  sleeve gastrectomy.

The case was rare and worthy of report, and authors were recommended to describe more in detail as below.

1. What devise did authors use at sleeve gastrectomy? and which size of staple for the resection?

Response 1: I am grateful to you for the insightful question. As indicated by your question, the selection of the stapler based on the thickness of the stomach wall will indeed have a significant impact on the leakage from the stapling line. In this case, for the initial stapling during sleeve gastrectomy, the Signia™ Stapling System with Endo GIA™ Black Reload was used, and for the remaining stapling, Endo GIA™ Purple Reload were used. (line 159–162)

I have included this information as an additional description in the paper. 2. Which method of the oral care did authors do preoperatively? Did authors use oral disinfectant?

Response 2: Generally, we do not perform oral disinfection during gastric surgery. Oral disinfection was not conducted in this case either, and this has been described in the paper. (line 167–170)

3. Authors were recommended the infectious route of A.odontolyticus of the case considering above points. 

Response 3: In response to your comment, we have described in the paper that the infection is believed to have occurred during the process of extracting the resected stomach into the extraperitoneal space, rather than through the aforementioned route.

Round 2

Reviewer 2 Report

Authors revised the article appropriately according to reviewer's comments.

Author Response

Response to Reviewer 2 Comments

I appreciate the valuable feedback from you.